# Enhancing early detection of neurological and developmental disorders and provision of intervention in low-resource settings in Uttar Pradesh, India: study protocol of the G.A.N.E.S.H. programme

Moreno Toldo,[1] Swami Varishthananda,[2] Christa Einspieler [ID],[3] Neeraj Tripathi,[1] Anshu Singh,[2] Surendra K Verma,[1] Kanchan Vishwakarma,[2] Dajie Zhang,[3,4,5] Agyeya Dwivedi,[1] Ritika Gupta,[2] Sanjay Karn,[2] Nirmal Kerketta,[1] Ram Narayan,[1] Karuna Nikam Singh,[6] Sumitra Rani,[2] Akanksha Singh,[2] Divyanshu Singh,[1] Krishna Pratap Singh,[2] Navin Singh,[1] Neeraj Singh,[1] Rishi Singh,[1] Shyam P Singh,[2] Rakesh Srivastava,[1] Sandeep Srivastava,[1] Sanjeev Srivastava,[2] Gopal Yadav,[1] Preeti Yadav,[1] Sheshnath Yadav,[2] Sujata Yadav,[2] Peter B Marschik[3,4,5]

Dedicated to the late Swami Varishthananda, Ramakrishna Mission Home of Service, Varanasi, India

MT, SV and CE contributed equally.

For numbered affiliations see end of article.

**Correspondence to**
Christa Einspieler;
christa.einspieler@medunigraz.at

## ABSTRACT

**Introduction** Around 9% of India's children under six are diagnosed with neurodevelopmental disorders. Low-resource, rural communities often lack programmes for early identification and intervention. The Prechtl General Movement Assessment (GMA) is regarded as the best clinical tool to predict cerebral palsy in infants <5 months. In addition, children with developmental delay, intellectual disabilities, late detected genetic disorders or autism spectrum disorder show abnormal general movements (GMs) during infancy. General Movement Assessment in Neonates for Early Identification and Intervention, Social Support and Health Awareness (G.A.N.E.S.H.) aims to (1) provide evidence as to whether community health workers can support the identification of infants at high-risk for neurological and developmental disorders and disabilities, (2) monitor further development in those infants and (3) initiate early and targeted intervention procedures.

**Methods** This 3-year observational cohort study will comprise at least 2000 infants born across four districts of Uttar Pradesh, India. Community health workers, certified for GMA, video record and assess the infants' GMs twice, that is, within 2 months after birth and at 3–5 months. In case of abnormal GMs and/or reduced MOSs, infants are further examined by a paediatrician and a neurologist. If necessary, early intervention strategies (treatment as usual) are introduced. After paediatric and neurodevelopmental assessments at 12–24 months, outcomes are categorised as normal or neurological/developmental disorders. Research objective (1): to relate the GMA to the outcome at 12–24 months. Research objective (2): to investigate the impact of predefined exposures. Research objective (3): to evaluate the interscorer agreement of GMA.

## Strengths and limitations of this study

► General Movement Assessment in Neonates for Early Identification and Intervention, Social Support and Health Awareness (G.A.N.E.S.H.) includes the application of a non-intrusive, highly reliable assessment tool (the Prechtl General Movement Assessment, GMA) to assist early identification of a high risk for cerebral palsy and other neurological and developmental disorders in a low-income geographical cohort.

► G.A.N.E.S.H. aims to provide evidence as to whether a community-based identification and intervention programme facilitates early detection of neurological and developmental disorders and disabilities.

► Measurement biases may occur because blinding is not possible: it has taken the medical doctors and other staff members involved 15 years to win the vulnerable population's trust, which is the basis for conducting a study as outlined here.

► Higher-order impacts may be underestimated as a follow-up period of 24 months may be too short because certain neurological and developmental disorders cannot be diagnosed with absolute certainty at this age.

► Developmental delay, in particular, may occur after weaning from breast feeding because of undernourishment, and might therefore not be detected by means of GMA during the first months of life.

**Ethics and dissemination** G.A.N.E.S.H. received ethics approval from the Indian Government Chief Medical Officers of Varanasi and Mirzapur and from the Ramakrishna Mission Home of Service in Varanasi. GMA

is a worldwide used diagnostic tool, approved by the Ethics Committee of the Medical University of Graz, Austria (27-388 ex 14/15). Apart from peer-reviewed publications, we are planning to deploy G.A.N.E.S.H. in other vulnerable settings.

## INTRODUCTION
### Background and problem context

India has the world's largest birth ratio (about 26 million per year) and is experiencing dramatic improvements in infant and child survival, although neonatal and infant mortality rates are still high at 30 and 41 per 1000 live births, respectively.[1] With a population of more than 199 million (ca. 16% of the national population), Uttar Pradesh is the most populous state in India. The 2011 census[2] showed that 59 million people in Uttar Pradesh live below the poverty line. The state's maternal mortality rate is the second highest in India with 258 maternal deaths per 100 000 live births. The under-five mortality rate is 90 per 1000 live births.[3] Lack of antenatal care, unsafe child births, undernourishment and lack of access to healthcare infrastructure are among the main risk factors for maternal and child morbidity and mortality. Neurological and developmental disorders and disabilities in particular compromise the attainment of full social and economic potential at family and community levels. The rate of occurrence of neurodevelopmental disorders ranges from 2.9% to 18.7% in 2-year to 9-year-old children across five Indian regions, and its all-site pooled estimate is 9.2% for children younger than 6 years.[4] In other words, almost one in 11 children has at least one of the following impairments: neuromuscular disorders and cerebral palsy, epilepsy, intellectual disability, speech and language disorders, hearing and/or vision impairment or autism.[4] Pooled estimates are unaffected by gender or rural/urban residence but increase by up to 3 percentage points when adjusted for national rates of low birth weight and/or stunting.[4] The overall pooled prevalence of cerebral palsy per 1000 children in India is 2.95 (95% CI 2.03 to 3.88).[5]

Apart from low birth weight (<2500 g) and preterm birth (<37 weeks' gestation), non-institutional delivery, a history of perinatal asphyxia (delayed cry and difficult breathing at birth), neonatal illness requiring hospitalisation, postnatal brain infections and stunting are significantly associated with neurological and developmental disorders and disabilities.[4 5] Several of these risk factors could be averted through public health intervention. There is strong evidence that early intervention improves functional outcomes for infants with neurological and developmental disorders and is economically cost-effective as it reduces the rate and severity of later impairments.[6 7] Early identification of neurodevelopmental impairment is therefore vital for ensuring timely referral to the limited resources provided for impoverished children. However, low-income communities, which are most vulnerable to neurological and developmental disorders, often lack programmes for early identification and intervention.[8]

Government programmes such as India's Integrated Child Developmental Services provide supplementary food, health and nutrition education, health check-ups, growth monitoring and child-specific services like immunisation. But to this day, only 20%–30% of people from marginalised groups access these services.[9] Non-governmental organisations, and especially their community health workers, are indispensable for improving the access to early identification and intervention programmes since they work hand in hand with Accredited Social Health Activists (ASHAs), Auxiliary Nurse Midwives (ANMs) and Anganwadi Workers, all of whom are part of India's public healthcare system. It is therefore of utmost importance that community health workers acquire relevant knowledge and skills to be able to recognise early markers for neurological and developmental disorders and refer infants in need to paediatric and neurological examinations. This goal has been supported by the Bill & Melinda Gates Foundation with a Grand Challenges Explorations grant for the development of smartphone-based applications as assistive tools.[10]

### About the organisations involved

The impact is assessed by two non-governmental organisations, Kiran Society (KS) and Ramakrishna Mission (RKM), in 10 blocks across four districts in the south east of Uttar Pradesh, India.

KS is a non-profit organisation situated near Varanasi (also known as Benares) on the banks of the river Ganges in Uttar Pradesh. Since 1990, KS has sought to empower marginalised groups and children with disabilities through rehabilitation, education, vocational training, social integration and advocacy (www.kiranvillage.org). KS offers institution-based and community-based rehabilitation services providing neurological examinations and interventions, physio-, occupational-, speech/language- and psychotherapy as well as corrective orthopaedic aids and appliances. Based on the WHO's community-based rehabilitation matrix and on three schemes, KS' community-based rehabilitation unit currently provides services for 200 000 people in the districts of Varanasi and Mirzapur. One of these schemes is the General Movement Assessment in Neonates for Early Identification and Intervention, Social Support and Health Awareness (https://kiranvillage.org/?page_id=3450): G.A.N.E.S.H. was launched in 2018.

RKM is a worldwide organisation founded in 1900. Its Varanasi branch, the RKM Home of Service, has a 230-bed hospital, nine indoor wards and 15 outpatient departments with various diagnostic units. Supported by the WHO, the European Commission, the British Medical Association and others, RKM Varanasi launched its Health Promotion Programme in 2000. Its main objectives are health education, primary healthcare services and training of community-based personnel in Varanasi and nearby districts. Its 15 actions and research projects include the following three, which are particularly relevant for G.A.N.E.S.H: (1) child healthcare education (esp.

multimedia presentations on feeding practices, hygiene and sanitation); (2) training community health workers and traditional birth attendants in promoting mother and child health and (3) mobile mother and child health clinics along with family planning and immunisation interventions. Since 2009, RKM has provided antenatal care and gynaecological services performed by mid-level women health workers in rural areas around Varanasi.

Both organisations' community health workers use mobile tools to pursue a broad range of health education goals including subjects such as vaccination, nutrition, hygiene and sanitation, but most importantly, they provide regular personal care. In addition, both organisations assist the most socioeconomically disadvantaged families (especially pregnant women and lactating mothers, infants and young children) in terms of food supplements containing vitamins, iron and calcium.

### Project overview and aims

The present observational study is being carried out in a population-based cohort of underprivileged infants in four districts of Uttar Pradesh, India. We are assessing the motor repertoire of individuals aged 1–5 months and its association with (1) demographic, prenatal and perinatal risk factors, and (2) the neurodevelopmental outcome in the second year of life. The study is conducted by the two non-governmental organisations portrayed above, which have long worked together and with governmental organisations in the target area. Disadvantaged families know that KS and RKM provide assistance in case of need.

The staff involved in both organisations received basic and advanced training in the Prechtl General Movement Assessment (GMA),[11] a diagnostic tool for early assessment of the integrity of the young nervous system. Training was provided by a licensed tutor of the GM Trust (CE), who also supervises the GMA and data evaluation during the project. Infants at high-risk for neurological and developmental disorders and disabilities are further examined by a paediatrician at RKM (SV) and a neurologist at KS (MT), who have worked together closely for more than 15 years. Another paediatrician is volunteering to assess infants and children in paediatric camps. All three of them are GMA-trained. KS has a special Medical Fund to guarantee that infants in need receive further diagnostic examination such as electroencephalography (EEG), X-ray, MRI or even genetic testing. If indicated, early physiotherapy is provided by KS staff, while both KS and RKM's teams provide basic health education and counselling.

We expect G.A.N.E.S.H. to improve health outcomes in children living in the target area. The main aims of this 3-year project are to (1) train community health workers to contribute to the identification of infants at high-risk for neurological and developmental disorders early on; (2) monitor further assessments in those infants and (3) initiate early and targeted intervention procedures (table 1, figure 1). Early identification is based on detailed assessments of the spontaneous motor behaviour[11] of all eligible infants born in eight Varanasi villages, 19 Mirzapur villages, one Sonbhadra cluster of villages and one Azamgarh cluster of villages. Research objective (1) is to relate the quantity and quality of movement and postural patterns—which will be determined by means of the Prechtl GMA[11] at 1–5 months—to the outcome at 12–24 months. Outcome will be assessed by paediatric and

| Table 1 | Main procedures and outcome measures |
|---------|--------------------------------------|
| **Procedures** | **Output** |
| Training community health workers to identify infants at high-risk for neurological and developmental disorders and disabilities early on | (1) Staff trained and certified in basic and advanced GMA<br><br>(2) Agreement (a) between the scorers, (b) between the scorers and a GMA-trained neurologist and (c) between the scorers and a GMA expert |
| Monitoring further assessments of infants at high-risk for neurological and developmental disorders and disabilities | (3) Number of infants screened<br><br>(4) Number of infants identified (abnormal age-specific GMs, reduced MOS)<br><br>(5) Number of families supported<br><br>(6) Number of underweight infants<br><br>(7) Number of infants examined by a paediatrician and/or a neurologist; examination results<br><br>(8) Number of (breastfeeding) mothers and infants who received micronutrient supply<br><br>(9) Number of infants who received (financially supported) access to MRI, EEG, X-ray and laboratory testing |
| Launching targeted intervention procedures early on | (10) Number of staff trained in early parent-oriented intervention<br><br>(11) Number of infants referred to early intervention<br><br>(12) Number of families referred to further parent training (around the infant's age of 6 to 7 months) |

EEG, electroencephalography; GMA, general movement assessment; MOS, motor optimality score.

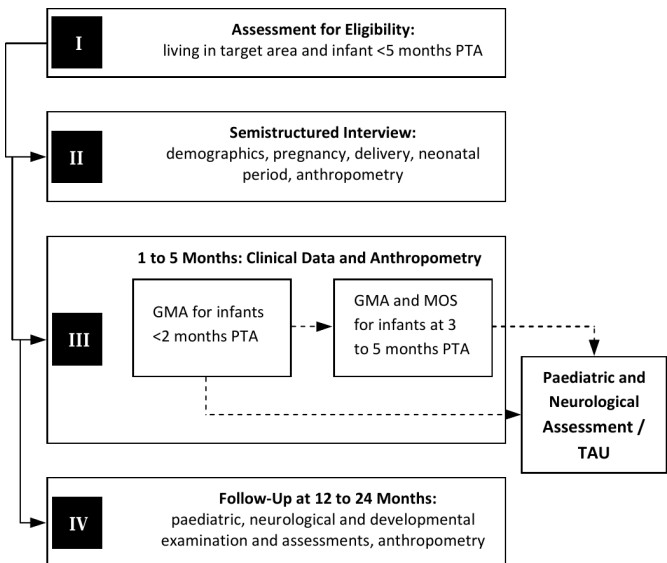

**Figure 1** Study design of the G.A.N.E.S.H. programme. GMA, general movement assessment; MOS, motor optimality score; PTA, post-term age; TAU, treatment as usual.

neurological examination (including the HINE)[12] as well as developmental testing (Trivandrum Developmental Screening Charts, TDSC).[13] Research objective (2) is to investigate whether a number of predefined exposures such as (1) the access to and frequency of antenatal care, (2) maternal micronutrient deficit (such as iodine, iron and vitamin A deficiency) during pregnancy according to the medical history in case of institutional delivery, (3) safety of child birth (home vs institutional delivery), (4) low birth weight (<2500 g) and undernourishment of the infant (weight, length and head circumference at the time of GMA being 2 SD below the reference group) and (5) intrapartum-related complications (difficulties to breathe and/or delayed cry at birth) are associated with movements and postures assessed at 1–5 months. Research objective (3) is to examine interscorer agreement between community health workers and GMA experts.

## METHODS AND ANALYSES
### Study design
Observational cohort study which follows the Strengthening the Reporting of Observational Studies in Epidemiology guidelines.

### Study setting and profile
In 2019, G.A.N.E.S.H. was implemented in 29 (cluster of) villages in 10 blocks of four districts of Uttar Pradesh, where the burden of poverty and undernutrition is particularly high: the Varanasi district (roughly 3 700 000 inhabitants; 1535 km$^2$; eight villages in the Kashi Vidyapeeth block, which is covered by KS); the Mirzapur district (roughly 2 500 000 inhabitants; 4521 km$^2$; 12 villages in the Majhawan and Shikhar blocks, likewise covered by KS; seven villages or cluster of villages in the Chhanbey,

Haliya, Jamalpur, Lalganj, Majhawan, Pahari and Patehara blocks, covered by RKM); the Sonbhadra district (roughly 1 800 000 inhabitants; 6788 km$^2$; one cluster of villages in the Robertsganj block, covered by RKM) and the Azamgarh district (roughly 4 600 000 inhabitants; 4054 km$^2$; one cluster of villages in the Lalganj block, covered by RKM).

Eastern Uttar Pradesh (particularly the districts of Mirzapur and Sonbhadra) has one of the lowest health indicators of India's eight socioeconomically backward states, also referred to as Empowered Action Group states. Poor health is characterised by lack of nutrition, chronic diseases and a high prevalence of infectious diseases resulting both in high mortality and morbidity and also in an increase of poverty due to out-of-pocket expenses and lost earnings.[14] The main sources of livelihood are agriculture, carpet weaving and pottery, but youth unemployment is high. According to the latest census from 2011,[2] the literacy rate then was between 64% in the Sonbhadra district and 75% in the Varanasi district, with a 15%–20% lower literacy rate in women.

The target area is characterised by a lack of trained medical staff willing to work in rural areas and a lack of healthcare infrastructure. Most of the care is provided by the family. Families who seek treatment will often turn to non-allopathic providers such as practitioners of Indian traditional medicine and quacks (ie, unqualified and unskilled health practitioners pretending to have medical skills).[14] The lack of neurodevelopmental health professionals, particularly in rural areas, places specialist neurodevelopmental care out of the reach of most people. The G.A.N.E.S.H. programme addresses this critical issue and seeks to provide healthcare to families in need.

The target population of G.A.N.E.S.H. comprises approximately 50 000 people. We aim to assess a cohort of at least 2000 infants born in the first and second years of the study (2019–2020) and collect their outcome data during the second and third years of the study (2020–2021). Due to the COVID-19 pandemic, data collection is deferred since 15 March 2020 and the timeline of the study will be rescheduled.

In order to discuss the procedure and aims of the G.A.N.E.S.H programme, community health workers of KS and RKM arranged meetings with ASHAs and ANMs in each block. ANMs are village-based female health workers establishing the first contact between the community and health services; in remote areas, ANMs also conduct home deliveries;[15] usually each ANM is supported by four to five ASHAs. ASHAs are local women trained as health educators and appointed by the Ministry of Health and Family Welfare of India. Their tasks include trying to persuade women to give birth in hospitals, taking children to immunisation clinics, encouraging family planning, treating basic illnesses and injuries with first aid and improving village sanitation.[15]

Each community health worker involved in the project is linked with one ANM and will be informed by them about each live birth in her village. Every Wednesday and

Saturday, the government offers free services in primary health centres in our target areas, where ANMs and ASHAs check the infants' weight and growth, promote and assist with breastfeeding and other nutritional aspects, and vaccinate. Community health workers linked to G.A.N.E.S.H. can also recruit mothers with newborn and young infants there. The community health worker approaches the parent(s) within 4 weeks after delivery and explains the G.A.N.E.S.H. project in detail, including, for example, mutual commitments. He/she hands out leaflets with relevant information provided in words (Hindi) and pictures. Parents/caregivers are assured of confidentiality and anonymity in reports and publications generated as part of G.A.N.E.S.H. Participation is voluntary, and families are assured that refusing to participate will not affect their access to care. If parents agree to participate, their (written) consent is obtained. Illiterate parents are informed in the presence of a literate witness and consent with their finger-print.

## Participants

The birth rate in the target area varies from 19.9 to 28.7 per 1000.[2] With a population of 50 000, we expect to see 1000–1500 newborns per year. Apart from living in the target area, inclusion criteria comprise all infants younger than completed 5 months of age irrespective of gender, family background, medical history and current health status. Infants older than 5 months of age are excluded.

Community health workers will collect demographic (please see also the Variables section), including the household composition and living condition, data from the mothers. Besides the medical history during pregnancy and delivery, maternal age and weight will also be documented.

The G.A.N.E.S.H. project is perceived by the communities as part of the ongoing services, which have been provided by KS and RKM for decades. More than 90% of newborn infants are brought to the free services in the primary health centres mentioned above, or, alternatively, the community health workers are welcomed at the family home. As part of the Indian culture, after having given birth, young mothers may move to the grandmother's home, possibly in another district. As the majority of the maternal grandparents also live in the same region, the number of moves to other regions is extremely small. Altogether, the number of families not willing to participate in the G.A.N.E.S.H. project is negligibly small.

## Variables

After the recruitment of the infant, a community health worker collects the following data (ie, exposure variables): demographical data (infant's gender, infant's age in days, parents' age, marital status, number of children, family member with disability, living status and household composition, religious affiliation and caste); maternal medical condition, especially during pregnancy and delivery, according to the documentation of antenatal care and/or the medical history collected during institutional delivery; maternal nutritional status (weight and height); access to and frequency of antenatal care; site of delivery (home vs institutional/hospital) indicating safety of child birth; complications during labour (documented in the medical history of the delivery) including intrapartum-related complications such as difficulties to breathe and/or delayed cry at birth; the infant's gestational age at birth and birth weight. Frequency of antenatal care, age, weight, height, number of children, parental age and maternal weight are continuous data; all other data are categorical.

At 1–5 months, we assess the following variables: (1) GMs: from birth to the end of the second month post-term age, we assess whether GMs are normal, poor repertoire or cramped-synchronised[11] (categorical data). From the beginning of the third month to the end of the fifth month post-term age, we assess fidgety movements as normal, abnormal or absent[16] (categorical data) and (2) the motor optimality score[17] (MOS; ordinal data) and its five subscores (ordinal data) as well as the number of normal and abnormal movement and postural patterns (continuous data); (3) clinical data include status of breastfeeding and the infant's medical condition since birth such as fever, diarrhoea (categorical data); (4) weight (measured with a portable weighing scale), length (measured on an infantometer) and head circumference (measured with a tape measure extending from the middle of the forehead to the farthest part in the rear of the head) are given as Z-scores, weight-for-age Z-scores and weight-for-length Z-scores (continuous data).

The outcome at 12–24 months is either normal or classified as neurological and developmental disorders including cerebral palsy, epilepsy (orthopaedic) congenital malformations, muscular dystrophy and spinal muscular atrophy, developmental delay, genetic disorders, visual impairment, hearing impairment, or a high risk for autism spectrum disorder (without the list being exhaustive; categorical data). The scores of the Hammersmith Infant Neurological Examination (HINE)[12] are ordinal data. In addition, each child's nutritional status is assessed and categorised as normal, underweight, wasted (weight-for-age Z-scores that are more than 2 SD below the median of the WHO child growth reference standards) or stunted (with height-for-age Z-scores that are more than 2 SD below the median of the WHO child growth reference standards). Head circumference Z-scores (continuous data) will also be documented.

## Data sources and measurement

All variables described in the first paragraph of the section 'Variables' are obtained by a community health worker abiding by a predefined interaction protocol and, if available, from the medical records of delivery.

Motor behaviour at 1–5 months is assessed by means of the Prechtl GMA,[11] which is widely regarded as the best clinical tool to predict cerebral palsy in infants under 5 months[18] and has proved invaluable for contributing to identify children at risk for developmental delay,[11]

cognitive dysfunction,[18] intellectual disabilities,[19] [20] speech and language disorders[21] and autism spectrum disorders.[22] GMA is a diagnostic tool for an early assessment of the integrity of the young nervous system based on observation of young infants' spontaneous movements. It is entirely non-intrusive and extremely time-efficient and cost-efficient. For a reliable assessment (kappa values ranging from 0.88 to 0.92),[11] [17] a 2–5-min video of the infant is recorded following a number of basic principles (infant lying in supine position, comfortably dressed, not crying).[11] Performed by trained assessors, the summary estimates for sensitivity and specificity of GMA are 0.98 (95% CI: 0.74 to 1.00) and 0.91 (95% CI: 0.83 to 0.93), respectively.[23] GMA is ideal for the neurological assessment of young infants, especially in low- and middle-income countries that struggle to provide expensive medical equipment for diagnoses.[24] From birth to the end of the second month post-term age, normal general movements (GMs) include a fluctuating sequence of arm, neck, trunk and leg movements. They wax and wane with changing intensity, speed and range of motion. Their onset and ending is typically gradual and smooth. Changing of direction and rotations across the trunk and limb axes are fluent and elegant.[11] From 3 to 5 months, GMs appear as fidgety movements with a very small amplitude and moderate speed of shoulders, wrists, hips and ankles in all directions and of variable acceleration.[11] Most infants will develop normally if fidgety movements are present and normal, especially if they co-occur with other smooth and fluent age-specific movement patterns.[11] [16] [18] The neural mechanisms of GMs are specific neural networks, central pattern generators (CPG), which are most probably located in the brain stem. We believe that the CPGs for GMs are constantly modulated by more rostral parts of the brain. The resulting variable movements in turn modulate the CPG-activity by variable sensory feedback.[11] [18] If a structural impairment of the young nervous system becomes functional, supraspinal projections are impaired and reduce the modulation of CPG activity, resulting in less variable GMs.[18] Before 3 months of age, abnormal GMs are described as: (1) 'poor repertoire' (the sequence of movements and the intensity, speed and range of motion lack variability) or (2) 'cramped-synchronised' (the limb and trunk muscles appear rigid and contract almost simultaneously, then relax almost simultaneously).[11] As a sign of functional impairment, fidgety movements can be (3) 'abnormal' (exaggerated in amplitude and speed), (4) sporadically present (not exceeding 3 seconds[25]) or (5) missing altogether at a post-term age of 9–16 weeks (6) and (7) are summarised as 'absent fidgety movements'.[11] [16] [26] Summary estimates of 0.98 for sensitivity and 0.91 for specificity make cramped-synchronised GMs followed by absent fidgety movements very reliable early markers of spastic cerebral palsy.[16] [17] [23] [25]

The motor repertoire of infants aged 3–5 months consists of fidgety movements as well as other movement patterns such as swiping, wiggling-oscillating limb movements,

kicking, arm and leg movements to the midline, fiddling, leg lifts or arching.[17] [27] The detailed GMA for 3-month to 5-month-old infants comprises the following five subcategories: (1) temporal organisation and quality of fidgety movements; (2) quality of other observed movement patterns; (3) age-adequate movement repertoire; (4) postural patterns and (5) movement character. Adding the scores of the five subcategories reveals the MOS[11] [17] with a maximum of 28 (ie, best possible performance) and a minimum of 5. An MOS ranging from 25 to 28 is considered to be optimal; scores from 20 to 24 are mildly reduced and an MOS below 20 requires intervention. A score below 9 indicates a very high risk for neurodevelopmental disabilities, especially for non-ambulatory cerebral palsy.[17] Various studies revealed that a reduced MOS was associated with later motor,[17] [28–30] cognitive[29] [31] and language[21] [29] impairments. A recent worldwide study on 468 children later diagnosed with cerebral palsy shows that such a detailed assessment of early movements and postures provides insight into a child's later functional capabilities and impairment.[17] Intraobserver reliability is high with intraclass correlation coefficients (ICC) ranging from 0.80 to 0.98[17] [25] and Cohen Kappa ranging from 0.87 to 0.91.[30]

In February and November 2018, KS and RKM staff involved in G.A.N.E.S.H. (medical doctors, physiotherapists, medical social workers, community-based rehabilitation field workers, special educators and one psychologist) received training from CE. They are now GM Trust-certified (basic and advanced level; www.general-movements-trust.info) to perform GMA. As a side note, for better readability, we have summarised the professions of medical social workers, community-based rehabilitation field workers and special educators under the term 'community health workers' throughout the manuscript.

After parental consent has been received, a GMA-trained KS/RKM staff member videotapes the infant's GMs according to GMA standards (the infant shows 2–5 min of active wakefulness, must not be crying, fussy or sucking on a pacifier, and must be lying in supine position without manipulation).[11] Video-recordings are preferably at two points in time: (1) during the first 2 months of life; and (2) during the fidgety movement period between the end of the second month and the end of the fifth month post-term age (corrected for preterm delivery). Infants are videotaped in the parents' home, in health camps, or in a primary health centre. The initial evaluation is performed by the GMA-certified staff member who has recorded the video, the second evaluation is performed by the first author (MT) and a third evaluation by CE, who is not familiar with the medical history of the infant. CE supervises the GMA as licensed tutor of the GM Trust. In a pilot sampling approach, we recorded and assessed 400 infants revealing ICCs ranging from 0.85 to 0.92 between the primary and secondary assessors and the supervisor.

The neurological assessment at 12–24 months follows the protocol of the HINE.[12] This method is easily performed and was designed for examining children

between 2 and 24 months of age. It includes 26 items that assess cranial nerve function, posture, quality and quantity of movements, muscle tone and reflexes and reactions. Each item is scored from 0 (minimum) to 3 (maximum score) revealing an overall global score from 0 to 78 (best neurological performance).[12] Both the sensitivity and the specificity are around 0.90.[32]

The TDSC[13] is based on 17 test items from the Bayley Scales of Infant Development (Baroda Norms). It was validated both at hospital and community levels and has a sensitivity of 0.85 and a specificity of 0.91, which makes it a simple, reliable screening tool that can be applied by community health workers.[33] The test-giver draws a vertical line through the chart indicating the actual age of the child. If the child can complete items that are to the left of the line, then there is no delay for that item. If an item lies to the left of the line and the child cannot complete this item, then an item delay is assumed. In this case, a trained assessor at KS will administrate the Bayley Scales of Infant and Toddler Development, Third Edition (cognition, language, motor, social–emotional and adaptive behaviour domains) with the child. Bayley-III scores exceeding 85 are considered to be normal. In addition, two psychologists (one at KS and one at RKM) were trained to identify sociocommunicative deficits which, along with repetitive and restricted behaviours, may indicate an increased likelihood of developing autism spectrum disorder.

A potential confounder especially after weaning from breastfeeding (usually after 6 months of age) might be undernutrition. It can be directly attributed to inadequate dietary intake, infection or disease that may affect the child. Lack of sanitation and hygiene, inadequate care, maternal mental health, economic deprivation and food insecurity might also be factors influencing the child's development. Undernutrition and other external disadvantages hampering thriving will be revealed and documented at all assessment appointments.

### Intervention programmes

The infant is referred to a targeted intervention programme if at least one of the following criteria is observed: (1) cramped-synchronised GMs during the first 2 months; in case of abnormal poor repertoire GMs during the first 2 months, GMA is repeated at 3–5 months, which will determine the further procedure; (2) absence of fidgety movements or exaggerated, abnormal fidgety movements at 3–5 months; (3) MOS <20; (4) asymmetries of any kind, especially asymmetric segmental wrist and/or finger movements;[17] (5) body weight and/or head circumference at or below the third age-related percentile on the WHO charts; (6) facial dysmorphism and/or malformations such as congenital talipes equinovarus, cleft palate; (7) acute illness and (8) any other parental concern.

Infants at high-risk for neurological and developmental disorders and disabilities and infants who need medical examination are referred to a paediatrician at RKM (SV)

and/or a neurologist at KS (MT), who have worked together closely for more than 15 years. A second paediatrician is volunteering to assess infants and children in paediatric camps. All three of them are GMA-trained. KS has a special Medical Fund to guarantee that infants in need receive further diagnostic examination such as EEG, X-ray, MRI or even genetic testing. If indicated, physiotherapist, occupational-therapist, speech/language therapists and special educators treat infants individually at home or at the Parents and Child Care Unit of KS. Since G.A.N.E.S.H. has been launched, every staff member involved in interventions has received regular workshops and practical demonstrations of early intervention techniques.

Trained RKM and KS personnel also offers education and counselling on basic health, nutrition and hygiene. Severely anaemic or malnourished mothers and underweight infants are given supplementary nutritious food, vitamins and micronutrients (especially calcium and vitamin D, iron and folic acid tablets) as well as medical care. Infants who need intensive medical treatment are referred to the Ramakrishna Hospital or the district hospitals of Varanasi and Mirzapur. Families in need are invited early on to attend paediatric or rehabilitation camps providing free services, examinations and, if necessary, additional medication.

### Bias

Measurement biases may occur because blinding is not possible: it has taken the medical doctors and staff members 15 years to win the participating population's trust, which is the basis for conducting this study. To avoid the interviewer bias, the community health worker will follow a predefined interaction protocol. In case the medical record of pregnancy and delivery is not available (lack of antenatal care or home delivery), a potential recall bias of the mother is hard to be ruled out. We will be able to control the confounders (eg, undernutrition), as we will document any recognisable external disadvantages hampering thriving, and compare the Z-values of the anthropometric measurements at birth, 1–5 months and 12–24 months.

### Study size

Taken an alpha cut-off of 5% and a beta cut-off of 20%, the sample shall involve 1261 children (ie, with a pooled incidence of 9.2% neurodevelopmental disorders in Indian children younger than 6 years,[4] and an expected incidence of 7% in the study group, with a birth rate in the target area varying from 19.9 to 28.7 per 1000,[2] that is, we expect to see 1000–1500 newborns per year, resulting in about 70–105 children with neurological and developmental disorders and disabilities) to achieve a predictive power of 0.95 and an effect-size of 0.15 using Cohen's $f^2$. According to our eligibility criteria, we aimed to assess a cohort of at least 2000 infants born in the first and second years of the study. Until 15 March 2020 (13 months after launching the study, before data collection was deferred

due to the COVID-19 pandemic), we have successfully recruited 1300 infants.

## Statistical methods

Data will be analysed according to the research objectives of the study using IBM SPSS Statistics in its updated version with significance set at $p < 0.05$. A lead biostatistician of the University Medical Center Goettingen, Germany, will provide expert input for the analyses.

Initial descriptive analysis will explore the characteristics and distributions of the variables. Any outlying data will be identified and the missing data explored. Histograms, boxplots and scatter plots, tables of means and SD or medians, interquartile ranges and cross tabulations will be used as appropriate.

Sensitivity, specificity and positive and negative predictive values will be determined with 95% CIs. This will be based on the positive/negative 'high risk of neurological and developmental disorders' status on GMs and MOS (at 3–5 months) compared with confirmed or suspected neurological and developmental disorders/no neurological and developmental disorders diagnoses (at 12–24 months). The association between the MOS (at 3–5 months) and the HINE (at 12–24 months) will be examined using Spearman's rank-order correlations. To test the predictive power of the MOS-subcategories (polytomous ordinal predictors) on the HINE cut-off outcomes (polytomous ordinal criterion), we will run ordinal regression analysis. The predictive power of abnormal movement and postural patterns (metric predictors) on HINE cut-off outcomes (polytomous ordinal criterion) will be again estimated by means of ordinal regression analysis.

We will perform regression analyses to determine how exposure variables are related to GMs and MOS. General linear model analyses will be performed to understand the effect and interactions of predefined variables and covariates on GMs. In case of multiple subgroup comparisons, we will divide the significant level of 0.05 by the total number of subgroups of the analysis (Bonferroni correction).

GMs and MOS will be rated by at least one community health worker, the first (MT) and the third author (CE). Fleiss' kappa will be calculated to assess the agreement among the three scorers when assigning categorical ratings to GMs. ICC statistics will be applied to examine the agreement of the MOS among the scorers.

## Longer-term outcomes

The listed longer-term outcomes are those we expect to achieve in a follow-up evaluation period of 3–5 years, depending on the availability of funds. Details will be determined later, as they can change according to requirements. Long-term outcomes should reflect an improved child health and nutritional status, improved motor, cognitive and language abilities, a better supply of vision and hearing aids and orthoses, proper management in case of epilepsy, access to MRI, EEG, X-ray and other diagnostic procedures, as well as adequate psychological

treatment and counselling—in other words: a proper management of neurological and developmental disorders and disabilities and a strengthened health system.

## Patient and public involvement

It has not been appropriate to involve infants and their families or the public in the design, or conduct, or reporting or dissemination plans of G.A.N.E.S.H.

## Ethics and dissemination

G.A.N.E.S.H. received ethics approval from the Indian Government Chief Medical Officers of Varanasi and Mirzapur and from the Ramakrishna Mission Home of Service in Varanasi. GMA is a worldwide used diagnostic tool, approved by the Ethics Committee of the Medical University of Graz, Austria (27-388 ex 14/15).

After parents/caregivers have received detailed information on G.A.N.E.S.H., they are assured of confidentiality and anonymity in reports and publications generated as part of G.A.N.E.S.H. Participation is voluntary, and families are assured that refusing to participate will not affect their access to care. If parents agree to participate, their (written) consent is obtained. Illiterate parents are informed in the presence of a literate witness and consent with their fingerprint. All study data are stored in secure databases. Paper files are kept in locked filing cabinets, and computer and video files are stored on password-protected servers. Only staff members associated with the G.A.N.E.S.H. programme have access to the data.

Results are submitted to peer-reviewed journals and presented at national and international conferences. Beyond that we aim for G.A.N.E.S.H. to serve as a model for early identification of and intervention in children with (or a high-risk for) neurological and developmental disorders and disabilities to have a sustainably impact on healthcare plans for low-income settings. Therefore, dissemination to healthcare professionals and policy makers is crucial.

All results presented are group data; individual participants are not identifiable. We intend to analyse and publish GMA and MOS results as well as 12-month to 24-month outcomes as standalone research articles in peer-reviewed journals. In accordance with our main aim to improve the quality and quantity of interactions with beneficiary families, the main research questions (in chronological order) to answer and discuss are as follows:

(1) Is it feasible for trained community health workers to launch G.A.N.E.S.H. and apply GMA in a rural, low-income setting? Key aspects: success of training, acceptance by ASHAs and ANMs, acceptance in the community, challenges, interscorer agreement, frequent staff meetings, expert supervision. Limitations/risks: parental refusal, cultural setting in case of male community health worker, migration of trained staff, priority of nutritional aspects over GMA.

(2) Does G.A.N.E.S.H. improve the frequency of home visits for infants and their mothers and the extent and level of counselling for families in need, and does it help

to identify abnormal motor behaviour and malnutrition? Key aspects: identification of abnormal GMs and reduced MOS, parental compliance with further paediatric and neurological examination, launching of targeted family-oriented intervention, targeted health education, provision of food supplementation and micronutrients. Limitations/risks: no compliance with further assessment and intervention because early markers are subtle; families see the need to consult a doctor only in case of (frequent) seizures, acute infection or malformations like cleft palate and congenital talipes equinovarus. The need for prevention and early intervention is difficult to grasp for parents, and is often impeded by cultural hierarchical family structure.

(3) Are we able to identify early on (especially at 3–5 months) children who will have an adverse outcome around 12–24 months of age? Key aspects: predictive value of GMA, sustainable family acceptance and compliance with targeted strategies of early intervention. Limitations/risks: the nutritional status, especially after weaning from breastfeeding (second half of the first year of life) might have a greater impact on the development than the integrity of the nervous system determined by GMA.

**Author affiliations**
[1]Department of Medical Rehabilitation, Kiran Society for Rehabilitation and Education of Children with Disabilities, Varanasi, India
[2]Department of Community Medicine, Ramakrishna Mission Home of Service, Varanasi, India
[3]Division of Phoniatrics, Research Unit iDN (Interdisciplinary Developmental Neuroscience), Medical University of Graz, Graz, Austria
[4]Department of Child and Adolescent Psychiatry and Psychotherapy, University Medical Center Göttingen, Gottingen, Germany
[5]Leibniz ScienceCampus Primate Cognition, Goettingen, Germany
[6]Joints n Motion (JnM) Academy, Mumbai, India

**Acknowledgements** The authors are grateful to Professor Akmer Mutlu, Hacettepe University Ankara, Turkey for providing teaching material to the physiotherapy unit at Kiran Society; to Mrs Sikha Sharma for basic insights into the principles of early intervention; to Mr. Amarendra Kumar and Mr Vijay Pal for their support in community-based rehabilitation; to Dr Ruchi Pandey for her assistance in paediatric camps; to Mrs. Sangeeta JK, founder and president of Kiran Society, for introducing the community-based rehabilitation approach among Kiran Society's activities and her ongoing support and inspiration and to Miha Tavcar (www.scriptophil.at) for copy-editing the article.

**Contributors** MT conceptualised G.A.N.E.S.H. and conceived the research questions; he is principle investigator and secured funding for the project; he prepared the study protocol with feedback from all collaborators; he provides second opinions for GMA and supervises and provides neurological interventions; he performs HINE and trained the physiotherapists in the same; he drafted the manuscript and approved the final manuscript as submitted. SV conceptualised G.A.N.E.S.H. and conceived the research questions; he is principle investigator and secured funding for the project; he prepared the study protocol with feedback from all collaborators; he supervised and provided paediatric interventions applied in G.A.N.E.S.H., he provided critical review of the first manuscript and the first revision. CE conceptualised G.A.N.E.S.H. and conceived the research questions; she is principle investigator and secured funding for the project; she prepared the study protocol with feedback from all collaborators; she developed the data base for the project; she trained all staff members in GMA, supervises GMA and provides third opinions. She drafted the manuscript, its revisions and approved the final manuscript as submitted. NT supervises the data collection under the guidance of investigators; he developed the data base for the project; he supervises and provides physiotherapeutic interventions; he performs HINE and approved the final manuscript as submitted. AnS supervises the data collection under the guidance of investigators; she developed the data base for the project; she supervises and provides health education including antenatal care; she performs GMA, provided critical review of the manuscript and approved the final manuscript as submitted. SKV developed the data base for the project and performs GMA and TDSC; he approved the final manuscript as submitted. KV supervises the data collection under the guidance of investigators; she performs GMA and provides health education including antenatal care; she approved the final manuscript as submitted. DZ advised on statistical design of the study, provided critical review of the first and second revisions and approved the final manuscript as submitted. AD provides physiotherapeutic intervention and health education; he performs HINE and TDSC; he approved the final manuscript as submitted. RG, SK, SR, KPS, SanjS, ShY and SuY perform GMA and provide health education; approved the final manuscript as submitted. NK provides physiotherapeutic intervention and health education, he performs HINE and TDSC, and approved the final manuscript as submitted. RN performs GMA and TDSC, and provides health education. He approved the final manuscript as submitted. KNS provided expert advice on early intervention in low-resource settings and supervises interventions applied in G.A.N.E.S.H.; she approved the final manuscript as submitted. AkS developed the data base for the project; she performs GMA and provides health education including antenatal care; she approved the final manuscript as submitted. DS provides physiotherapeutic intervention and health education; he performs HINE and TDSC. He approved the final manuscript as submitted. NaS provides hearing assessments and coordinates physiotherapeutic interventions. He approved the final manuscript as submitted. NeS provides physiotherapeutic intervention and health education; he performs HINE and TDSC, and approved the final manuscript as submitted. RiS performs GMA and provides health education; she approved the final manuscript as submitted. SPS performs GMA and provides health education; he approved the final manuscript as submitted. RaS he supervises and provides physiotherapeutic interventions and performs HINE; he approved the final manuscript as submitted. SandS performs GMA and TDSC; he provides special education and approved the final manuscript as submitted. PY performs GMA and TDSC and provides health education; approved the final manuscript as submitted. GY performs GMA, HINE and TDSC; he approved the final manuscript as submitted. PBM is co-PI; he secured funding for the project, assisted in conceptualisation and prepared the study protocol with feedback from all collaborators; he provided critical review of the manuscript and approved the final manuscript as submitted.

**Funding** This work is supported by Bill & Melinda Gates Foundation (opp1128871), the Austrian Science Fund FWF (P25241, TCS24, KLI811), the GM Trust (WHEP23-8-5-16), the Australian Cerebral Palsy Alliance (SEG00419), the Old Possum's Practical Trust, the Medical Fund of Kiran Society and the Ramakrishna Mission.

**Competing interests** The GM Trust supports G.A.N.E.S.H. CE is a GM Trust tutor for GMA, and PBM is president of the GM Trust (honorary function), both of whom have contributed to G.A.N.E.S.H. on an honorary basis.

**Patient consent for publication** Not required.

**Provenance and peer review** Not commissioned; externally peer reviewed.

**ORCID iD**
Christa Einspieler http://orcid.org/0000-0002-7875-0632

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
