## [Reviewer comments · BMJ Open]

ARTICLE DETAILS

TITLE (PROVISIONAL)	Enhancing early detection of neurological and developmental disorders and provision of intervention in low-resource settings in Uttar Pradesh, India: Study protocol of the G.A.N.E.S.H. programme
AUTHORS	Toldo, Moreno; Varishthananda, Swami; Einspieler, Christa; Tripathi, Neeraj; Singh, Anshu; Verma, Surendra; Vishwakarma, Kanchan; Zhang, Dajie; Dwivedi, Agyeya; Gupta, Ritika; Karn, Sanjay; Kerketta, Nirmal; Narayan, Ram; Nikam Singh, Karuna; Rani, Sumitra; Singh, Akanksha; Singh, Divyanshu; Singh, Krishna; Singh, Navin; Singh, Neeraj; Singh, Rishi; Singh, Shyam; Srivastava, Rakesh; Srivastava, Sandeep; Srivastava, Sanjeev; Yadav, Gopal; Yadav, Preeti; Yadav, Sheshnath; Yadav, Sujata; Marschik, Peter

VERSION 1 – REVIEW

REVIEWER	Charles A. Nelson Boston Children's Hospital and Harvard Medical School USA
REVIEW RETURNED	30-Mar-2020

GENERAL COMMENTS	In high-income countries there is a long history of attempting to predict neurodevelopmental disorders in early childhood from measures obtained in infancy; this dates back to the development of the Apgar score and in the 1950s and 60s, in the United States, the large National Collaborative Perinatal Project that was particularly focused on predicting cerebral palsy. Admittedly the approach adopted in a high-income country might not generalize to a low-income setting, such as rural India. Nevertheless, it is not apparent to me that the authors are fully aware of this long history of research. I am particularly concerned about the emphasis on motor findings in the current project. I believe the evidence that supports the link between early motor findings and later disability is weak except for children who end up with several handicapping conditions (e.g., severe CP). Moreover, there is an abundance of evidence that early motor findings rarely associate with later autism outcomes except in a minority of cases. Additional concerns about this project is whether there are other tools that can be used in this setting that the authors are not considering (for example, see work coming out of the INS in Dhaka, Bangladesh); that the statistical approach is rather simplistic; and that no mention is made of how to account for missing data. There is one additional issue I feel compelled to raise which has to do with the ethics of screening at a population level. Specifically, I understand that children can be referred out for a more formal evaluation (e.g., by a child neurologist) but will services be readily available for children who receive a diagnosis of a disability/disorder? How long will the family have to wait for services? These issues should be considered in the context of the
---

	overall aims of the project. These limitations, notwithstanding, there are numerous strengths to this project, including: a) it is population-based; b) the use of community workers to conduct the screening exams; c) the project starts early in life (1-5 months); d) the implementation plan seems sound; e) there appears to be an adequate number of workers; f) this approach may in particular be sensitive to CP and epilepsy
--	---

REVIEWER	DR. KAI JIN Usher Institute, University of Edinburgh
REVIEW RETURNED	24-Jun-2020

GENERAL COMMENTS	This study aimed to detect the neurodevelopmental disorders (NDD) in infants in India. A prospective observational cohort study is used, which is appropriate design to examine the outcome interest (NDD) and exposure variables before the development of the disease. Such design is usually used to answer the question about the association between “risk factors” and disease outcome (neurodevelopmental disorders) and have the strength of temporal relationship between an exposure and an outcome. However, this paper requires for significant revision in research questions, methods and analysis. Lack of clarity of the research questions has impact on the research design and data analysis. I strongly recommend the authors follow the STROBE protocol (Strengthening the Reporting of Observational Studies in Epidemiology) particularly in the method section. Research questions  1. Objectives have been repetitively stated several times over the paper without the consistency:  1) Page 6 /26 , line 24 to 33; 2) Page 7/26 , line 3 to 30; 3) Page 13 -14; 4) aims were not stated in the abstract; 2. Lack of clarity:  1) Primary objective: “Our primary objective is to relate the quantity and quality of movement and postural patterns” What relationship is to be examined with the quantity and quality of movement and postural patterns? 2) Secondary objective “Our secondary objective is to relate the results to a number of predefined confounders. Specific questions will be whether (i) movements and postures at 1 to 5 months, (ii) the 12- to 24-month-outcome, and (iii) the correlation between (i) and (ii) are associated with (a) the access to and frequency of antenatal care, (b) a maternal micronutrient deficit during pregnancy, (c) the safety of child birth,(d) a low birth weight (< 2500g) and undernourishment of the infant, and (e) a delayed cry at birth.” What are “the results” to be examined? Are these cofounders or exposures?
--

	Methods  1. Study design:  a. The schematic of study design would be very helpful to understand this design and strongly recommended to provide; 2. Study population:  a. Give the eligibility criteria to include and exclude criteria; b. Describe the sampling methods: e.g. convenient sampling; c. Describe method of follow-up. What is the efforts to improve the future loss to follow up; 3. Page 12 intervention program: not clear, please clarify:  a. Do the infants with abnormal repertoire GM at 3-5months receive the intervention? b. How long is the intervention; c. Are outcome of interest assessed at 12 -24 month after intervention? d. Any intention to compare the effectiveness of the intervention in this program? 4. Variables description:  a. Define the exposures ; b. Clearly defined primary outcome and secondary outcomes instead of using primary/secondary research aims; c. What are the confounding factors? Have the confounding factors taken account of the design and analysis? 5. Describe power and sample calculation and rationale Statistical methods: In principle, statistical analysis plan should reflect the research design and research questions, and emphasis should be on which analyses, comparisons, and statistical tests have been planned given the objective of the study.  1. Statistical analysis Page 14: "First, we evaluate descriptive parameters (medians and interquartile ranges or percentages) to summarise sample characteristics" What variables are to be summarised? How this analysis is relevant to the objectives of this study?  2. Page 15; "Odds ratios and 95% confidence intervals are calculated for (a) pre- and perinatal data that reveal a significant difference between normal and
--	---

	abnormal GMs, and (b) normal and abnormal GMs that reveal a significant difference between the categorical outcome variables assessed at 12 to 24 months.” Which research questions is this analysis for? 3. “To determine whether two or more independent samples had the same distribution in MOS and/or HINE” What is “two or more independent sample”, how is this relevant to the research questions? 4. What is the attempts within analysis plan to address for potential confounder? 5. Describe any methods used to examine subgroups and interaction 6. If there is loss to follow up, how to address the missing data 7. Any plan for sensitive analysis
--	---

VERSION 1 – AUTHOR RESPONSE

Reviewer 1

In high-income countries there is a long history of attempting to predict neurodevelopmental disorders in early childhood from measures obtained in infancy; this dates back to the development of the Apgar score and in the 1950s and 60s, in the United States, the large National Collaborative Perinatal Project that was particularly focused on predicting cerebral palsy. Admittedly the approach adopted in a high-income country might not generalize to a low-income setting, such as rural India. Nevertheless, it is not apparent to me that the authors are fully aware of this long history of research.

REPLY

Thank you very much for the valuable comments to help amending our manuscript. We are very grateful for the critical remarks on our study protocol and the understanding of the need to adapt clinical and study procedures to the situation in Uttar Pradesh, India.

We are well aware of the long history of attempting to predict neurodevelopmental disorders in early childhood but refrained from reviewing the various attempts from Virginia Apgar to brain imaging in infants born preterm in high-resource settings in this protocol paper. The corresponding author (C.E.) has been working on this topic since the early 1990s.

We do agree with the Reviewer, that the approach adopted in a high-income country cannot be easily generalized to a low-income setting, such as the region of our focus in this

project, the rural Uttar Pradesh. Uttar Pradesh is the most populous state in India with 59 million people living below the poverty line, with a disability rate of 15.5%.

In 1997, *The Lancet* announced our easy to apply, non-invasive and non-intrusive General Movements Assessment (GMA) as the most promising diagnostic tool to identify infants at risks for neurological/developmental disorders and disabilities already at preterm, term, and early post-term age^{16 (Manuscript)}. Since then, more than 150 studies worldwide in high- and low-resource settings and several meta-analyses have repeatedly demonstrated the validity of this highly reliable and efficient predictive tool. In 2017, the American Medical Association published the Guidelines for early identification of an increased risk for cerebral palsy^{18 (Manuscript)} (with C.E. as co-author), stating “the 3 tools with best predictive validity for detecting cerebral palsy before 5 months’ corrected age are (1) neonatal magnetic resonance imaging (MRI) (86%-89% sensitivity), (2) the Prechtl Qualitative Assessment of General Movements (GMs) (98% sensitivity), and (3) the Hammersmith Infant Neurological Examination (HINE) (90% sensitivity)”^{18, page 899}. In 2018, Kwong et al.³⁴ systematically reviewed 47 studies and concluded (page 480) that “The Prechtl General Movements Assessment during the fidgety period (10–20wks corrected age) had the strongest sensitivity: 97 per cent (95% confidence interval [CI] 93–99) and specificity: 89% (95% CI 83–93). The sensitivity and specificity of the Prechtl GMA during the writhing period (birth–6wks) was 93% (95% CI 86–96) and 59% (95% CI 45–71) respectively. Cramped-synchronized movements in the writhing period according to Prechtl had the best specificity (sensitivity: 70% [95% CI 54–82]; specificity: 97% [95% CI 74–100]).”^{34, page 480}

Already in the Manual for the General Movement Assessment published as Clinics in Developmental Medicine Nr. 167,^{11 (Manuscript)} we emphasized that GMA should be combined with imaging techniques. A decade later, a systematic review on assessment techniques to predict cerebral palsy, revealed equally high predictive values for GMA and MRI^{24 (Manuscript)}. Several following studies also presented the high sensitivities and specificities of GMA for neurological and developmental disorders, both combined with or contrasted to MRI.^{e.g.35-39} These results lead to an ever-increasing number of health professionals around the world using GMA for early identification of infants at high risk for neurological and developmental disorders and disabilities. Especially in low and middle income countries where MRI is not available at all times and in all places, observation comes into play.^{17,25 (Manuscript)} Apart from studies in progress in Mexico, Brazil, Zimbabwe, Uganda, South Africa, Eastern Europe, Turkey, Iran, Kazakhstan, India, and rural China on early identification of neurological and developmental disorders, GMA is also applied to longitudinal cohorts of infants vertically exposed to Malaria,^{25 (Manuscript)} Tuberculosis, Chikungunya, Syphilis, CMV (all in progress), HIV,⁴⁰ or Zika Virus^{29 (Manuscript),41}. which adds to the confidence of the families.

Finally, we would like to highlight the distribution of neurological and developmental disorders and disabilities in 2019 in the target area of the project. Among 1,271 children treated in the outreach camps and at the Parent and Child Care Unit of the Kiran Society (the NGO working for the

G.A.N.E.S.H. project; https://kiranvillage.org/?page_id=3450), the majority (75%) were diagnosed with motor and orthopaedic disorders and problems (the primary diagnosis of the child), such as (in descending order): cerebral palsy (including epilepsy), post-polio residual paralysis, orthopaedic deformations and congenital talipes equinovarus, motor delay, muscular dystrophy, spina bifida, spinal muscular atrophy, spinal cord injury, and various genetic disorders affecting the motor system. Another 12% of the children were diagnosed with intellectual disabilities (including Down syndrome, which does not require the GMA for diagnosis). 7% of the children were diagnosed with hearing or visual impairment, and 4% with epilepsy. Only 2% were diagnosed with psychiatric disorders (ASD, ADHD, and psychosis). Although these numbers refer to the year 2019, the distribution remained similar during the last seven years. Highlighting ASD in the abstract and outlet might have been misleading; we have accordingly changed that in the manuscript.

Reviewer 1

Additional concerns about this project is whether there are other tools that can be used in this setting that the authors are not considering (for example, see work coming out of the INS in Dhaka, Bangladesh); that the statistical approach is rather simplistic; and that no mention is made of how to account for missing data.

REPLY

We appreciate the MRI studies carried out in Dhaka under the supervision of the Reviewer.^{e.g.44,45} We were impressed to read that the resting state networks appear to reflect differences between infants from the impoverished families and more affluent families.⁴⁵ We recently found that young infants with aberrant general movements exhibited decreased resting-state functional connectivity between the basal ganglia and regions in parietal and frontotemporal lobes.⁴⁶ This study was conducted at the University of Chicago Comer Children's Hospital; there is currently no chance to implement a similar study in Varanasi, India.

Both organizations involved in our G.A.N.E.S.H. project are non-profit, non-governmental organizations. Their activities are depending on funds and donations. MRI is not yet affordable for the general screening process. In case that GMA reveals abnormalities, and subsequent detailed neurological examination by a child neurologist (first author; having been working in the region since 15 years) confirms the necessity of a brain imaging, the infant will be referred to a private Diagnostic Centre (cooperating with our project initiators), or to the Banaras Hindu University, Department of Radio Diagnosis & Imaging, to perform a 1.5 Tesla MRI diagnostic scan.

We were able to acquire extra funding for additional statistical consultancy and have discussed our protocol with clinical advisors and clinical psychologists. We are convinced that state-of-the-art approaches will be applied once data collection is completed. To ensure statistical and mathematical correctness we will include the institutional statistic board at the University Medical Center Göttingen, Germany.

The G.A.N.E.S.H. project is perceived by the communities as part of the ongoing services, which have been provided by the Kiran Society and the Ramakrishna Mission for decades. Each community health worker involved in the G.A.N.E.S.H. project is linked with one Auxiliary Nurse Midwife (ANM) and will be informed of each live birth in the ANM's village. Twice a week, the government offers free services in primary health centres in our target areas, where ANMs and Accredited Social Health Activists (ASHAs) check the infants' weight and growth, promote and assist with breastfeeding and other nutritional aspects, and vaccinate. ANMs and ASHAs are a critical part of India's public health care system. The community health workers linked to G.A.N.E.S.H. recruit mothers with newborns and young infants at the primary health centres, or visit the infants at their home.

The first two authors (a child neurologist and a paediatrician) have both been working more than 15 years in the target area and are well known to the community. Approximately 90% of newborn and young infants are brought to the services mentioned above, or, alternatively, the community health workers are welcomed at the family home. The remaining 10% have moved to the mother's home (in another district) after having given birth. Though this is part of the Indian culture, the number of moves to other regions is small as the majority of maternal grandparents live in the same region. Altogether, the number of families not willing to participate in the G.A.N.E.S.H. project is negligibly small. All infants receive neurological and paediatric assessment and follow-up examination for free; micronutrient and, in case of need, food supply for free; in case of need of further evaluations such as affordable or free x-ray, surgical corrections (cleft palate, clubfoot), MRI, or genetic testing; free access to all therapies including 1- to 2-week camps for parent's training.

We have currently collected data of 1,300 families; 17 (1.3%) refused to participate in follow-up assessments and services; one infant with spinal muscular atrophy died. Hence, *loss to follow-up* is hardly present, and we do not expect any significant bias on the results.

Missing data mainly relate to birth weight. Especially after home-deliveries not all neonates are weighed. However, we take the anthropometry (Z-scores) in all infants at the appointment for the general movement video recording, which is at least once between postnatal day 4 to 120.

Reviewer 1

There is one additional issue I feel compelled to raise which has to do with the ethics of screening at a population level. Specifically, I understand that children can be referred out for a more formal evaluation (e.g., by a child neurologist) but will services be readily available for children who receive a diagnosis of a disability/disorder? How long will the family have to wait for services? These issues should be considered in the context of the overall aims of the project.

REPLY

Thank you very much for raising this important question. The screening (recording and assessment of general movements) is done by a community health worker certified for GMA.

Assessment is done within the first three days after video recording, mostly on the same day. In case of (a) abnormal general movements, (b) low weight, (c) diarrhoea, (d) fever, (e) exanthema or any other health conditions, and most of all, in case of (f) parental concerns that mainly relate to their observation of suspected fits, an appointment with the child neurologist or the paediatrician is scheduled within maximally 1 week. If indicated, EEG will be performed at the Hospital of the Ramakrishna Mission Home of Service. MRI is usually performed within 2 weeks after the prescription. In case the family cannot afford the transportation into the city of Varanasi, a driver from Kiran Society or Ramakrishna Mission will take care of the transportation. Community health workers are also assisting if the family members are not comfortable to bring the child to the expert for further evaluation.

Reviewer 1

These limitations, notwithstanding, there are numerous strengths to this project, including: a) it is population-based; b) the use of community workers to conduct the screening exams; c) the project starts early in life (1-5 months); d) the implementation plan seems sound; e) there appears to be an adequate number of workers; f) this approach may in particular be sensitive to CP and epilepsy.

REPLY

We are most grateful to the Reviewer for recognizing the strengths of our project. Indeed, the majority of infants already screened are diagnosed with motor impairments, often cerebral palsy, and epilepsy. Especially in case of fits observed by parents, an immediate referral to EEG (for which G.A.N.E.S.H. covers the costs) and the start of antiepileptic medication is imperative.

Thank you very much, indeed, for the comments and suggestions to amend our protocol paper. We hope to have addressed all issues properly.

Reviewer 2

This study aimed to detect the neurodevelopmental disorders (NDD) in infants in India. A prospective observational cohort study is used, which is appropriate design to examine the outcome interest (NDD) and exposure variables before the development of the disease. Such design is usually used to answer the question about the association between “risk factors” and disease outcome (neurodevelopmental disorders) and have the strength of temporal relationship between an exposure and an outcome.

However, this paper requires for significant revision in research questions, methods and analysis. Lack of clarity of the research questions has impact on the research design and data analysis. I strongly recommend the authors follow the STROBE protocol (Strengthening the Reporting of Observational Studies in Epidemiology) particularly in the method section.

REPLY

We are very grateful for the valuable comments and critical remarks on our study protocol. These were extremely helpful to revise and amend our manuscript. As suggested by the reviewer, the STROBE protocol is now implemented in this version, particularly in the Method section.

Reviewer 2

Research questions

1. Objectives have been repetitively stated several times over the paper without the consistency: 1) Page 6 /26 , line 24 to 33; 2) Page 7/26 , line 3 to 30; 3) Page 13 -14; 4) aims were not stated in the abstract;

REPLY

Thank you very much for pointing this out. Rewriting the manuscript according to the STROBE protocol we also double checked the manuscript for consistency (1-3 above; Abstract and end of Introduction section). Aims (4 above) have been added to the Abstract.

Reviewer 2

2. Lack of clarity:

1) Primary objective: “Our primary objective is to relate the quantity and quality of movement and postural patterns”. What relationship is to be examined with the quantity and quality of movement and postural patterns?

REPLY

The “quantity and quality of movement and postural patterns” refers to standard assessments of newborn neuromotor functions. To clarify, the relevant sentence reads as follows: “Our primary objective [now: research objective 1] is to relate the quantity and quality of movement and postural patterns assessed by means of the Prechtl GMA (at 1 to 5 months) and MOS (at 3 to 5 months) to the outcome at 12 to 24 months, which is assessed in paediatric and neurological examinations as well as developmental tests.”

We have checked similar aspects in a number of papers that associated assessments in infancy with outcome measures and found comparable phrasing in these publications. Indeed, there is a huge body of knowledge from all over the world^{e.g.17,25 (Manuscript)} that GMA (assessing movements and postures during early infancy as in the proposed protocol) has high predictive values for outcome assessments performed during toddlerhood,^{e.g.29 (Manuscript)} preschool age,^{e.g.47,48} school age,^{e.g.49,50} and even puberty⁵¹. However, none of these studies are carried out in low-resource settings.

Reviewer 2

2) Secondary objective “Our secondary objective is to relate the results to a number of predefined confounders. Specific questions will be whether (i) movements and postures at 1 to 5 months, (ii) the 12- to 24-month-outcome, and (iii) the correlation between (i) and (ii) are associated with (a) the access to and frequency of antenatal care, (b) a

maternal micronutrient deficit during pregnancy, (c) the safety of child birth, (d) a low birth weight (< 2500g) and undernourishment of the infant, and (e) a delayed cry at birth. "What are "the results" to be examined? Are these cofounders or exposures?

REPLY

We were indeed wrong to use the term "predefined confounders" instead of "exposure variables" for the factors listed from (a) to (e). Thank you for helping us to clarify this issue.

Reviewer 2

Methods

1. Study design:

a. The schematic of study design would be very helpful to understand this design and strongly recommended to provide;

REPLY

We are grateful for this suggestion and have added Figure 1 illustrating the study design.

Reviewer 2

2. Study population:

a. Give the eligibility criteria to include and exclude criteria;

REPLY

The eligibility criteria are now given in the Participants topic. We include infants younger than 5 months of age irrespective of gender, family background, medical history, and current health status. Infants older than completed 5 months are not included in the study.

Reviewer 2

b. Describe the sampling methods: e.g. convenient sampling;

REPLY

Our study is a population based study aiming to answer the research objectives for all infants younger than 5 months of age and living in a defined area, namely in 29 (cluster of) villages in ten blocks of four districts of Uttar Pradesh as described in the Setting section. We have defined a study interval of 2-years for the first assessments (1 to 5 months), which now needs to be expanded because of COVID-19 pandemic regulations.

Please see also reply below.

Reviewer 2

c. Describe method of follow-up. What is the efforts to improve the future loss to follow up;

REPLY

The G.A.N.E.S.H. project is perceived by the communities as part of the ongoing services, which have been provided by the Kiran Society and the Ramakrishna Mission for decades. The first

two authors (a child neurologist and a paediatrician) have both been working more than 15 years in the target area and are well known to the community. Approximately 90% of newborns and young infants are brought to the services or, alternatively, the community health workers are welcomed at the family home. The remaining 10% have moved to the mother's home (in another district) after having given birth. Though this is part of the Indian culture, the number of moves to other regions is small as the majority of maternal grandparents live in the same region. Altogether, the number of families not willing to participate in the G.A.N.E.S.H. project is negligibly small. Since many years, all infants receive neurological and paediatric assessment and follow-up examination for free. In addition, we provide micronutrient and food supply if needed; in case of need of further evaluations affordable or free X-ray, MRI, surgical corrections (cleft palate, clubfoot), or genetic testing; free access to all therapies including 1- to 2-week camps for parent's training.

We have currently collected data of 1,300 families; only 17 (1.3%) refused to participate in follow-up assessments and services; one infant with spinal muscular atrophy died. Hence, loss to follow-up is hardly present, and we do not expect any bias on the results.

But, of course, again the regulations due to COVID-19 pandemic might increase loss to follow-up and can hardly be foreseen now. Overall, the established child and family care through the involved institutions and the personal involvement of a sustainable health-care provider system guarantees that the loss to follow-up is reduced to a minimum.

Reviewer 2

3. Page 12 intervention program: not clear, please clarify:

a. Do the infants with abnormal repertoire GM at 3-5months receive the intervention?

REPLY

We have now added that infants with abnormal poor repertoire GMs during the first two months, will be re-checked concerning their GMs at 3 to 5 months and the result will determine the further procedure. As described in tops (b) and (c) of the Intervention topic in the main text, intervention is provided if the infant does not develop fidgety movements or if his/her fidgety movements are exaggerated, abnormal, and/or if the MOS <20.

Reviewer 2

b. How long is the intervention;

REPLY

As we are not studying the effect of intervention, and we are subsuming allopathic treatments (e.g. antiepileptic drugs), micronutrient supply, education in hygiene and sanitation, physio- and occupational therapy etc. as "intervention", treatment is subject to personalized medicine to meet the needs of the individual infant and his/her family.

Reviewer 2

c. Are outcome of interest assessed at 12 -24 month after intervention?

REPLY

In this study we are not evaluating the effectiveness of interventions (please see also answer below). As in numerous studies in high-resource settings, we are studying GMA as a predictor for outcome and thus related to diagnosis made.

Reviewer 2

d. Any intention to compare the effectiveness of the intervention in this program?

REPLY

Infants and families receive individual treatments tailored to their specific needs. We do not apply any special type of intervention which we could compare to the treatment as usual approach. Please keep in mind the very low resource setting and restricted possibilities in this respect. It is not an intervention study but rather aims at providing the best available treatment at the earliest possible time. It would certainly be of interest to evaluate different interventional approaches to this population but this is beyond the scope of the project and far beyond feasibility in such a vulnerable setting.

Reviewer 2

4. Variables description:

a. Define the exposures ;

REPLY

Following STROBE, we have now included a Variables topic where we defined all exposures accordingly.

Reviewer 2

b. Clearly defined primary outcome and secondary outcomes instead of using primary/secondary research aims;

REPLY

As we do not study the effect of any intervention, we feel the definition of outcome of greatest importance vs. evaluating additional effects of the intervention does not apply.

Reviewer 2

c. What are the confounding factors? Have the confounding factors taken account of the design and analysis?

REPLY

We have now included in the Variables topic that a potential confounder might be undernutrition, especially after weaning from breastfeeding after 6 months of age. It can be directly attributed to inadequate dietary intake or infection or disease that affects the child. Lack of sanitation and hygiene, inadequate care and maternal mental health, economic deprivation, and food insecurity

might be contributory factors. We will be able to control the confounder bias (undernutrition), as we will document any recognizable external disadvantages hampering thriving, and compare the Z-values of the anthropometric measurements at birth, 1 to 5 months, and 12 to 24 months.

Reviewer 2

5. Describe power and sample calculation and rationale

REPLY

We have now included a Study size topic stating the following: Taken an alpha cut-off of 5% and a beta cut-off of 20%, the sample shall involve 1,261 children (i.e. with a pooled incidence of 9.2% neurological/developmental disorders in Indian children younger than 6 years, and an expected incidence of 7% in the study group, with a birth rate in the target area varying from 19.9 to 28.7 per 1,000 we expect to see 1,000 to 1,500 newborns per year, resulting in about 70 to 105 children with neurological/developmental disorders and disabilities) to achieve a predictive power of .95 and an effect-size of .15 using Cohen's f^2 . According to our eligibility criteria, we aimed to assess a cohort of at least 2,000 infants born in the first and second year of the study. Until March 15, 2020 (13 months after launching the study, before data collection was deferred due to the COVID-19 pandemic), we have successfully recruited 1,300 infants.

Reviewer 2

Statistical methods:

In principle, statistical analysis plan should reflect the research design and research questions, and emphasis should be on which analyses, comparisons, and statistical tests have been planned given the objective of the study.

REPLY

With the assistance of a biostatistician, we have rewritten the Statistic methods topic according to the STROBE guidelines.

Reviewer 2

1. Statistical analysis Page 14: "First, we evaluate descriptive parameters (medians and interquartile ranges or percentages) to summarise sample characteristics" What variables are to be summarised? How this analysis is relevant to the objectives of this study?

REPLY

The reviewer is right; we did not summarise variables. What we meant is to describe the clinical characteristics of all infants involved in the study. We have rephrased the relevant sentences.

Reviewer 2

2. Page 15; "Odds ratios and 95% confidence intervals are calculated for (a) pre- and perinatal data that reveal a significant difference between normal and abnormal GMs, and (b) normal

and abnormal GMs that reveal a significant difference between the categorical outcome variables assessed at 12 to 24 months.”

Which research questions is this analysis for?

REPLY

With the assistance of a biostatistician, we have rewritten the Statistic methods topic according to the STROBE guidelines.

Reviewer 2

3. “To determine whether two or more independent samples had the same distribution in MOS and/or HINE”. What is “two or more independent sample”, how is this relevant to the research questions?

REPLY

With the assistance of a biostatistician, we have rewritten the Statistic methods topic according to the STROBE guidelines.

Reviewer 2

4. What is the attempts within analysis plan to address for potential confounder?

REPLY

Whether indeed undernutrition will be a confounder can be determined by comparing the Z-values of the anthropometric measurements carried out at three time points: birth, 1 to 5 months, and 12 to 24 months. The potential confounder(s) will be taken as covariates in our mixed ANOVA analyses.

Reviewer 2

5. Describe any methods used to examine subgroups and interaction

REPLY

As we are not conducting a clinical trial we did not describe subgroup analysis and interaction tests. In addition, previous studies^{e.g. 16,17,20,29 (Manuscript)} revealed that there is no subgroup (demographics, i.e. gender, recording age group) effect on the GMA identifying infants at high-risk for developmental disorders and disabilities. Following the advice of the reviewer, we shall certainly evaluate for any subgroup modification of the predictive power of the outcome. Our statistician shall conduct a test for interaction using GLM models to evaluate subgroup differences.

Reviewer 2

6. If there is loss to follow up, how to address the missing data

REPLY

We have currently collected data of 1,300 families; 17 (1.3%) refused to participate in follow-up assessments and services; one infant with spinal muscular atrophy died. Hence, loss to follow-up is hardly present, and we do not expect a significant bias on the results.

Missing data mainly relate to birth weight. Especially after home-deliveries not all neonates are weighed. However, we take the anthropometry (Z-scores) in all infants at the appointment for the general movement video recording, which is at least once between postnatal day 4 to 120.

Thank you very much for the valuable comments and suggestions to amend our protocol paper. We hope to have addressed all issues properly.

Additional References

34. Kwong AKL, Fitzgerald TL, Doyle LW, *et al.* Predictive validity of spontaneous early infant movement for later cerebral palsy: a systematic review. *Dev Med Child Neurol* 2018;60:480-489.
35. Maeda T, Iwata H, Sekiguchi K, *et al.* The association between brain morphological development and the quality of general movements. *Brain Dev* 2019;41:490-500.
36. Kelly CE, Thompson DK, Cheong JLY, *et al.* Brain structure and neurological and behavioural functioning in infants born preterm. *Dev Med Child Neurol* 2019;61:820-831.
37. Eeles AL, Walsh JM, Olsen JE, *et al.* Continuum of neurobehaviour and its associations with brain MRI in infants born preterm. *BMJ Paediatr Open* 2017;1:e000136.
38. George JM, Fiori S, Fripp J, *et al.* Relationship between very early brain structure and neuromotor, neurological and neurobehavioral function in infants born <31 weeks gestational age. *Early Hum Dev* 2018;117:74-82.
39. Peyton C, Yang E, Msall ME, *et al.* White Matter Injury and General Movements in High-Risk Preterm Infants. *AJNR Am J Neuroradiol* 2017;38:162-169.
40. Palchik AB, Einspieler C, Evstafeyeva IV, *et al.* Intra-uterine exposure to maternal opiate abuse and HIV: the impact on the developing nervous system. *Early Hum Dev* 2013;89:229-235.
41. Einspieler C, Marschik PB. The developmental spectrum of prenatal Zika virus exposure. *Lancet Child Adolesc Health* 2020;4:345-346.
42. Barnes F, Graham L, Loganathan P, *et al.* General movement assessment predicts neuro-developmental outcome in very low birth weight infants at two years - a five-year observational study. *Indian J Pediatr* 2020; doi: 10.1007/s12098-020-03365-1.
43. Peyton C, Einspieler C. General movements: a behavioral biomarker of later motor and cognitive dysfunction in NICU graduates. *Pediatr Ann* 2018;47:e159-e164.

44. Turesky T, Xie W, Kumar S, *et al.* Relating anthropometric indicators to brain structure in 2-month-old Bangladeshi infants growing up in poverty: A pilot study. *Neuroimage* 2020;210:116540.
45. Turesky TK, Jensen SKG, Yu X, *et al.* The relationship between biological and psychosocial risk factors and resting state functional connectivity in 2-month-old Bangladeshi infants: a feasibility and pilot study. *Dev Sci* 2019;22: e12841.
46. Peyton C, Einspieler C, Fjørtoft T, *et al.* Correlates of normal and abnormal general movements in infancy and long-term neurodevelopment of preterm infants: insights from functional connectivity studies at term equivalence. *J Clin Med* 2020;9:834.
47. Spittle AJ, Spencer-Smith MM, Cheong JL, *et al.* General movements in very preterm children and neurodevelopment at 2 and 4 years. *Pediatrics* 2013;132:e452-8.
48. Ferrari T, Todeschini A, Guidotti I, *et al.* General movements in full-term infants with perinatal asphyxia are related to basal ganglia and thalamic lesions. *J Pediatr* 2011;158:904-911.
49. Roze E, Meijer L, Van Braeckel KN, *et al.* Developmental trajectories from birth to school age in healthy term-born children. *Pediatrics* 2010;126:e1134-e1142.
50. Fjørtoft T, Grunewaldt KH, Løhaugen GC, *et al.* Adaptive behavior in 10-11 year old children born preterm with a very low birth weight (VLBW). *Eur J Paediatr Neurol* 2015;19:162-169.
51. Einspieler C, Marschik PB, Milioti S, *et al.* Are Abnormal Fidgety Movements an Early Marker for Complex Minor Neurological Dysfunction at Puberty? *Early Hum Dev* 2007;83:521-525.

VERSION 2 – REVIEW

REVIEWER	Charles A. Nelson Boston Children's Hospital and Harvard Medical School Harvard Graduate School of Education USA
REVIEW RETURNED	07-Aug-2020
GENERAL COMMENTS	The authors have done an excellent job responding to the previous review.
REVIEWER	KAI JIN University of Edinburgh
REVIEW RETURNED	03-Aug-2020
GENERAL COMMENTS	Thanks very much for authors' detail responds. The authors have addressed the comments well. Some issues require for clarifications: 1. 16/61: Participants should also include the infant's mothers --- infant's mothers data are essential part of this study and analysis and being collected (e.g, mother's age, SES B/G , maternal medical conditions).

	2. 13/61 Research objective 2 is to investigate whether a number of predefined exposures such as (a) the access to and frequency of antenatal care, (b) maternal micronutrient deficit during pregnancy, (c) safety of child birth, (d) low birth weight (<2500g) and undernourishment of the infant, and (e) delayed cry at birth are associated with movements and postures assessed at 1 to 5 months. a. Providing the definition of the outcomes or how these outcomes were measured: e.g. maternal micronutrient deficit during pregnancy, safety of childbirth, undernourishment of the infant, delayed cry in the outcome measurements. 3. 17/61: Data source and measurement: would recommend providing a list (or in a table format) for data collected in the semi-structure interview and medical records of delivery for easy reading. 4. Page 17/6, line 51-53: "Exposure variables are obtained through a semi-structured interview and from the medical records of delivery. " Line 52, suggest removing "exposure", 5. Thanks for the authors for providing STROBE statement which has improved transparency for the readers to follow. It would be fine to left in the supplementary file.
--	---

VERSION 2 – AUTHOR RESPONSE

Reply to Reviewer 2

Reviewer 2

Thanks very much for authors' detail responds. The authors have addressed the comments well.

Some issues require for clarifications:

1. 16/61: Participants should also include the infant's mothers --- infant's mothers data are essential part of this study and analysis and being collected (e.g, mother's age, SES B/G , maternal medical conditions).

REPLY: We have added this information to the Participants section. Thank you very much again for your valuable comments and suggestions.

Reviewer 2:

2. 13/61 Research objective 2 is to investigate whether a number of predefined exposures such as (a) the access to and frequency of antenatal care, (b) maternal micronutrient deficit during pregnancy, (c) safety of child birth, (d) low birth weight (<2500g) and undernourishment of the infant, and (e) delayed cry at birth are associated with movements and postures assessed at 1 to 5 months.

a. Providing the definition of the outcomes or how these outcomes were measured: e.g. maternal micronutrient deficit during pregnancy, safety of childbirth, undernourishment of the infant, delayed cry in the outcome measurements.

REPLY: We have now added how the exposures mentioned in Research Objective 2 were measured.

Reviewer 2:

3. 17/61: Data source and measurement: would recommend providing a list (or in a table format) for data collected in the semi-structure interview and medical records of delivery for easy reading.

REPLY: We have rephrased "semi-structured interview" into "interaction protocol" and described the (exposure) variables obtained by the community health workers in the Variable section.

Reviewer 2:

4. Page 17/6, line 51-53: "Exposure variables are obtained through a semi-structured interview and from the medical records of delivery. "

Line 52, suggest removing “exposure”,

REPLY: We have deleted “exposure” and rephrased this sentence.

Reviewer 2:

5. Thanks for the authors for providing STROBE statement which has improved transparency for the readers to follow. It would be fine to left in the supplementary file.

REPLY: Thank you very much for the critical review of our manuscript and the suggestions made.